# Knowledge, Attitude, and Behaviour with Regard to Medication Errors in Intravenous Therapy: A Cross-Cultural Pilot Study

**DOI:** 10.3390/healthcare11030436

**Published:** 2023-02-03

**Authors:** Noemi Giannetta, Meysam Rahmani Katigri, Tahere Talebi Azadboni, Rosario Caruso, Gloria Liquori, Sara Dionisi, Aurora De Leo, Emanuele Di Simone, Gennaro Rocco, Alessandro Stievano, Giovanni Battista Orsi, Christian Napoli, Marco Di Muzio

**Affiliations:** 1School of Nursing, UniCamillus-Saint Camillus International University of Health and Medical Sciences, 00131 Rome, Italy; 2Social Determinants of Health Research Center, Saveh University of Medical Sciences, Saveh, Iran; 3Health Information Management, Iran University of Medical Sciences, Tehran, Iran; 4Health Professions Research and Development Unit, IRCCS Policlinico San Donato, San Donato Milanese, 20097 Milano, Italy; 5Department of Biomedical Sciences for Health, University of Milan, 20133 Milano, Italy; 6Department of Biomedicine and Prevention, University of Rome Tor Vergata, 00133 Rome, Italy; 7Nursing, Technical, Rehabilitation, Assistance and Research Department, IRCCS Istituti Fisioterapici Ospitalieri—IFO, 00144 Rome, Italy; 8Center for Excellence in Nursing Scholarship, OPI, 00173 Rome, Italy; 9Department of Clinical and Experimental Medicine, University of Messina, 98125 Messina, Italy; 10Department of Public Health and Infectious Diseases, Sapienza University of Rome, 00185 Rome, Italy; 11Department of Surgical and Medical Sciences and Translational Medicine, Sapienza University of Rome, 00185 Rome, Italy; 12Department of Clinical and Molecular Medicine, Sapienza University of Rome, 00185 Rome, Italy

**Keywords:** medication errors, cross cultural comparison, nursing, intensive care units

## Abstract

Background: Literature on the prevention of medication errors is growing, highlighting that knowledge, attitude and behavior with regard to medication errors are strategic to planning of educational activities and evaluating their impact on professional practice. In this context, the present pilot study aims to translate and validate nursing professionals’ knowledge, attitudes and behavior (KAB theory) concerning medication administration errors in ICU from English into Persian. Furthermore, two main objectives of the project were: performing a pilot study among Iranian nurses using the translated questionnaire and carrying out a cultural measurement of the KAB theory concerning medication administration errors in an ICU questionnaire across two groups of Italian and Iranian populations. Methods: A cross-cultural adaptation of an instrument, according to the Checklist for reporting of survey studies (CROSS), was performed. The convenience sample was made up of 529 Iranian and Italian registered nurses working in ICU. An exploratory factor analysis was performed and reliability was assessed. A multi-group confirmatory factor analysis was conducted to test the measurement invariance. Ethical approval was obtained. Results: There was an excellent internal consistency for the 19-item scale. Results regarding factorial invariance showed that the nursing population from Italy and Iran used the same cognitive framework to conceptualize the prevention of medication errors. Conclusions: Findings from this preliminary translation and cross-cultural validation confirm that the questionnaire is a reliable and valid instrument within Persian healthcare settings. Moreover, these findings suggest that Italian and Persian nurses used an identical cognitive framework or mental model when thinking about medication errors prevention. The paper not only provides, for the first time, a validated instrument to evaluate the KAB theory in Iran, but it should promote other researchers in extending this kind of research, supporting those countries where attention to medical error is still increasing.

## 1. Introduction

The National Coordinating Council for Medication Error Reporting and Prevention defines a Medication error as any preventable event that may cause or lead to inappropriate medication use or patient harm while the medication is in the control of the health care professional, patient, or consumer [1]. Such events may be related to professional practice, health care products, procedures and systems, including prescribing, order communication, product labelling, packaging and nomenclature, compounding, dispensing, distribution, administration, education, monitoring and use [1]. Literature on medication errors prevention is growing, demonstrating that these events may occur at any stage of the medication process and in different healthcare settings, such as general hospitals but also primary care, elderly acute care and pediatric facilities [2,3,4,5,6,7,8,9]. Specifically, the rate of medication errors in Intensive Care Units (ICUs) is higher than all other settings due to the ICU environment (e.g., staff workload) and the nature of patients admitted to ICUs [10,11]. However, due to differences in measures for medication error rates, to this date the rate of medication errors is not completely known.

Furthermore, several studies have investigated the impact of medication errors from a nurse perspective and barriers to error reporting by nurses [12,13,14,15,16]. The results highlight that knowledge, attitude and behavior with regard to medication errors are strategic for the planning of educational activities and the evaluation of their impact on professional practice. Although increasing, the literature focusing on nursing professionals’ knowledge, attitude and behaviour (KAB theory) in the preparation and administration of medication does not cover the majority of the countries [17].

KAB theory regarding medication administration errors led to an ICU questionnaire which was recently developed to investigate these factors in the nursing population [17]. This questionnaire has been developed to measure, knowledge, attitudes and behaviour with regard to the preparation and administration of intra-venous (IV) drugs. It also encompassed participants’ socio-demographic and professional characteristics, access to up-to-date information and the need for continuous education related to the use of IV medication. The questionnaire was used in the Italian nurse population in 2017 and subsequently it was adapted to the Spanish context by Márquez-Hernández et al. in 2019 [18], to the Maltese context in 2020 [19] and to homecare settings in 2021 [20]. Although the questionnaire has been translated into different languages for use in different cultural groups, it has never been cross-culturally validated to ensure that scores, across different cultural clusters, could be meaningfully related [21]. In fact, according to Thoma et al. [22], the construct of medication errors, their incidence and etiological factors need also to be analyzed with regards to the personnel’s cultural background. For instance, some studies report common features in the distribution and type of medication errors between the Italian and Iranian contexts [12,13,14,15,16,17,23,24,25].

According to Mansouri et al. [24], in the Iranian context from 29.8% to 47.8% of medication errors occur during the prescribing stage; from 10.0% to 51.8% of medication errors occur during transcribing, from 11.3% to 33.6% of medication errors occur during dispensing and from 14.3% to 70% of medication errors occur during the administration stage. Medication errors in the administrative stage are the most common medical in clinical settings (53%). In the Italian context, 16.5% of medication errors occur during the prescribing stage. 11% of medication errors occur during transcribing and 13.5% of medication errors occurs during the dispensing stage [25].

Although data are consistent between the two countries, the analysis of the contexts in which the studies were conducted reveals some important differences that need to be investigated. This observation is confirmed in a recent systematic literature review, which highlights the need to analyze the construct of medication errors—their incidence and etiological factors—in relation to cultural background [22].

In this context, the present study aims to translate and validate the questionnaire regarding the use of KAB theory in the study of medication administration errors in ICUs from English into Persian. Furthermore, two main objectives of the project are performing a pilot study among Iranian nurses using the translated questionnaire and carrying out a cultural measurement of KAB in medication administration errors in ICUs across two groups of Italian and Iranian populations.

## 2. Materials and Methods

### 2.1. Study Design

A cross-sectional pilot study was performed to provide a validated tool to investigate the KAB theory in medication administration errors in ICUs, involving ICU nurses and carrying out cultural measurement of KAB in a questionnaire on medication administration errors in ICUs between two groups of an Italian and Iranian population. Reporting was evaluated by the Checklist for reporting of survey studies (CROSS) (Appendix A).

### 2.2. Iranian Sample

The sample was a convenience sample. Overall, all the Persian registered nurses who work in ICUs in the hospital in Tehran, Iran (n.44) completed a self-administered questionnaire. Researchers sent an email to all nurses working in ICU. Consent was obtained before starting the data collection. Based on a web survey, data was collected in February 2019. The inclusion criteria were nurses who worked in the ICU at the hospital hosting the research who could speak and read Persian fluently.

As Italian control group, the 529 nurse sample reported by Di Muzio et al. was used [17]. In this first Italian study, the value of Cronbach’s alpha of the questionnaire resulting from 19 items was 0.776 [17]. The questionnaire was used also in other contexts, such as Spanish and Maltese hospitals [18,19].

The questionnaire also investigates the participants’ degree of agreement with each item, using a scale that ranges from 1 (Totally disagree) to 5 (Totally agree). However, the statements in the section on attitudes were provided with a three-level Likert scale (Agree, Uncertain and Disagree).

### 2.3. Persian Validation Study

The English version of the KAB medication administration errors in ICU questionnaire was used as a starting point in the translation process. In this process, the authors used the recommendations written by the World Health Organization [26] and Beaton et al. [27]. First, two Iranian investigators translated the English versions of the instrument into Persian (Appendix A). To confirm if the Persian translations were reliable, an expert translated them back into English. Afterward, a panel of two experts (fluent in English and Persian) evaluated and compared the retroversion with the original English version and confirmed their accuracy.

### 2.4. Data Analysis

Data obtained from the questionnaire administration was extracted on an Excel worksheet and associated with alphanumeric variables to identify the items and their responses. Subsequently, statistical analysis of the variables was carried out through SPSS^®^ statistical software for macOS, version 25. Participants’ socio-demographic and professional characteristics were presented using descriptive statistics. Internal consistency was evaluated using Cronbach’s alpha. Usually, Cronbach’s alpha ≥ 0.70 is considered a satisfactory internal consistency level [28,29,30].

#### 2.4.1. Exploratory Factor Analysis

In order to understand if there was an adequate sample and an appropriate data structure to conduct an exploratory factor analysis (EFA), the Bartlett’s test of sphericity and the Kaiser-Meyer-Olkin (KMO) were used. According to Williams et al. [28], the *p*-value should be less than 0.05 and the KMO value should exceed 0.50.

Construct validity was evaluated using principal axis factoring with orthogonal rotation of extracted factors by varimax rotation (Kaiser normalization). Eigenvalues greater than 1.0 and a scree graph were used as general criteria.

#### 2.4.2. Confirmatory Factor Analysis

The measurement equivalence was searched based on the Meredith framework [31]. Indeed, multi-group confirmatory factor analysis was conducted to evaluate the configural, metric, scalar and strict invariance of the scale across the two populations. To evaluate model fitting [32], the following goodness-of-fit indices were assessed: chi-square (χ2) statistics; comparative fit index (CFI) and Tucker and Lewis index (TLI); root mean square error of approximation (RMSEA); and standardized root mean square residual (SRMR). CFI or TLI ranging from 0.90 to 0.95 were considered acceptable values [33,34]. An RMSEA or SRMR value lower than 0.08 confirms a moderate fit [35,36,37]. These statistical analyses were carried out through the Mplus^®^, version 7. To verify the configural, metric, scalar and strict invariance of the scale across the two populations, we compared the fit of nested models using the chi-square difference tests (Δχ2). According to Cheung & Rensvold [38], we considered a significant change in the fit model if ΔCFI > 0.01.

### 2.5. Ethical Considerations

Ethical approval was obtained from the Institutional Ethical and Review Boards of Saveh University of Medical Sciences (IR.SAVEHUMS.REC.1399.026). Participation in the study was voluntary and anonymous. Participants were asked to compile the questionnaire. Informed consent was obtained from all subjects involved in the study.

## 3. Results

### 3.1. Characteristics of the Sample

#### Professional and Socio-Demographic Characteristics of the Iranian Sample

Table 1 shows the demographic and professional characteristics of the nursing sample. The majority of the Iranian sample (79.5%) was female. The average age across the Iranian nursing sample was 36.52 years. All nurses in the sample had obtained a university degree. Regarding post-basic qualifications, 6.8% (n = 3) of the sample stated that they had graduated with a master’s degree, and 93.2% (n = 41) had attended “Other”. The average work experience among the nursing sample was about 11 years. Most of the sample had attended courses on medication management during their university degree and post-degree courses (n = 25; 56.8%). The sample showed a good level of English language knowledge (n = 22; 50.0%), had internet at work (n = 39; 88.6%) and did not have a library at work (n = 23; 52.3%). Most of the nursing sample (n = 38; 86.4%) spent less than 1 h per week improving learning. Data concerning the Italian control group were acquired from a previous paper by Di Muzio et al. [17] and summarized in Table 1.

### 3.2. Exploratory Factor Analysis

Bartlett’s test of sphericity was significant (χ2(171) = 881.160; *p* < 0.001) and the Kaiser-Meyer-Olkin (KMO) measure of sampling adequacy was 0.862. According to these results, an exploratory factor analysis (EFA) was run.

Table 2 shows the results of the EFA analysis. Factors were defined by the level of association of the variables found in the analysis of factor load and their subjectivity. The initial solution identified a three-factor solution, with 58.26% of the variance explained. The scree plot identified one primary factor and levelled out after three (Figure 1).

Principal axis factoring with orthogonal rotation of extracted factors by varimax rotation (Kaiser normalization) was used. After rotation, eight items were kept by factor 1; factor 2 predicted seven items and, factor three was loaded with four items. Individual item factor weightings are detailed in Table 2.

This means that the instrument can be divided into three sections, i.e., Knowledge, attitude and behaviour, according to the original research [17].

### 3.3. Internal Consistency

There was internal consistency for the 19-item scale (Cronbach’s alpha = 0.951).

Knowledge of the medication errors prevention scale of the instrument demonstrated a Cronbach’s alpha of 0.925 (8 items). The attitude to the medication errors prevention scale of the instrument demonstrated a Cronbach’s alpha of 0.932 (7 items). The behaviour medication errors prevention scale of the instrument demonstrated a Cronbach’s alpha of 0.947 (4 items). Table 2 shows the Item-Total Statistics that presents the Cronbach’s Alpha if ‘Item Deleted’.

### 3.4. Confirmatory Factor Analysis and Measurement Equivalence

#### 3.4.1. Measurement Equivalence of Knowledge Scale

Firstly, we tested the configural invariance: the two groups (Italian and Iranian nursing) have the same cognitive framework for conceptualization of medication errors prevention. As shown in Table 3, the multi-sample knowledge model showed the following goodness-of-fit indices: χ2(28) = 39.552, *p* < 0.001; RMSEA = 0.094; CFI = 0.939; χ2 contribution: 14.285 (Italian nurses) and 25.464 (Iranian nurses). The study of loadings reveals important aspects: there is a negative loading regarding the first item in the group of Italian nurses, not statistically significant (*p* = 0.903), and not confirmed in the group of Iranian nurses. Except for the loadings of item 5 in the group of Italians (also not statistically significant), the remaining loadings are all positive and statistically significant. In other words, configural invariance has been verified, i.e., the same number of factors and the same cognitive framework are present in both groups. In addition, metric invariance was assessed. The comparison between the first model and the latter showed a Δχ2 equal to 13.447 (Δgl = 7), not significant (*p* = 0.06), and a ΔCFI = −0.03. These results suggest that the first-order factors of the knowledge section of the questionnaire possess the same measurement scale in the two samples. Therefore, metric invariance can be considered to have been achieved. Table 3 also reports the study results for scalar invariance and strict invariance.

#### 3.4.2. Measurement Equivalence of the Attitude Scale

Measurement equivalence of the Attitude scale was tested in the two samples: configural invariance is verified, with discrete goodness-of-fit indices: χ2 (35) = 79.116, *p* < 0.001; RMSEA = 0.079; CFI = 0.904; χ2 contribution: 40.594 (Italian nurses) and 38.522 (Iranian nurses). Again, the loadings of the items were all positive and statistically significant. Table 3 also reports the results of the scalar invariance and the strict invariance study.

#### 3.4.3. Measurement Equivalence of Behaviour Scale

Configural invariance and metric invariance were achieved when comparing model fits between the Italian and Iranian samples in the behaviour scale. Specifically, the goodness-of-fit indices are as follows: χ2 (10) = 17.519, *p* = 0.06; CFI = 0.979; contribution of χ2: 9.409 (Italian nurses) and 8.110 (Iranian nurses). Metric invariance also appears to be confirmed with a Δχ2 of 0.732 (Δgl = 5), non-significant (*p* = 0.98) and a ΔCFI = −0.01. Thus, these results show metric equivalence, i.e., the 5-step Likert response scale is used almost identically by both populations (Italian and Iranian) (Table 3).

## 4. Discussion

Medication errors (MEs) are a global concern all over the world and can occur in all kind of medication [39]. Several studies have explored MEs and adherence to preventive measures in all healthcare settings and also at environmental level [40,41,42]. However, assessing KAB about MEs among nurses is strategically important for the provision of an evidence-grounded framework to guide interventions for dealing with this global threat. This study provides evidence of the validity and reliability of a self-report measure of KAB about MEs in the nursing population in Italy and Iran. In particular, the results show that the same cognitive framework characterizes the questionnaire validated in this research in order to conceptualize MEs prevention. Factors that might lead to MEs are related to healthcare workers’ organizational variables or human characteristics. KAB of each healthcare worker is essential in medication errors prevention. KAB theory with regard to medication administration errors in ICU questionnaire is an instrument validated in 2018, aimed to evaluate the three factors of KAB theory in MEs prevention among the nursing population. This study aimed to translate, cross-culturally adapt, and test the psychometric properties of the KAB theory in the medication administration errors, using an ICU questionnaire in the Persian language.

The factors identified in our study (Knowledge about MEs prevention, Attitude to MEs prevention, Behaviour regarding MEs prevention) demonstrate some alignment with those identified by Di Muzio et al. [17], Márquez-Hernández et al. [18], and Giannetta et al. [19]. The instrument demonstrated an excellent internal consistency for the 19-item scale in the Persian context. This findings concerning internal consistency are in line with those reported in the Spanish context [18], but they do not confirm the results of another study in an English-speaking context [19].

Indeed, in the Spanish study carried out using the same tool, the reliability of the instrument, assessed using Cronbach’s Alpha, was 0.849 The value of the KMO test was 0.867 and the Bartlett sphericity test was significant (χ2(171) = 2146.118; *p* < 0.05), indicating that the analysis of the data factors was appropriate. However, in the Maltese version, the value of the KMO test was 0.931 and the Bartlett sphericity test was significant, but the reliability of the instrument, using Cronbach’s Alpha, was 0.626 [19].

In the first validation study, the questionnaire also reported a lower value for Cronbach’s alpha resulting from 19 items [17], than in the present one. Analysis for redundancy indicated that Cronbach’s alpha would not increase if items were removed.

The mean for knowledge scores about MEs prevention, which were all greater than 4, indicated excellent knowledge of the skills needed to prevent MEs. The mean for knowledge scores about MEs prevention was higher in the Italian and Maltese studies [17,19]. In this Iranian study, the lower score for the knowledge scale was due to the dosage calculus for intravenous drugs, even if its value was greater than 4.

According to these findings, the Persian sample obtained higher values in the attitude to MEs prevention scale. The Italian and Maltese samples were all lower. These findings are in line with Thoma et al. [22], which affirm that the construct of MEs, their incidence and etiological factors need also to be analyzed with regard to the personnel’s cultural background.

However, this study showed that there is no difference in cognitive framework or mental model when nurses responded to the survey in very different countries, according to the multi-group confirmatory factor analysis. This is an important point because, as suggest by Mansouri [12], there is a limited number of studies in the literature in Iran. Based on this, the instrument suggested, and translated in this study into the Persian language, could be used to explore MEs prevention in two different contexts and the results from this study could be compared with other studies using the same instrument. Moreover, this survey allows for the simultaneous analysis of knowledge, attitudes and behaviors, which, according to Di Muzio, are closely interrelated. In fact, appropriate knowledges influence correct behavior, and positive attitudes also have influence on correct knowledges and behavior. This pilot study shows that even in the Persian population, different in cultural characteristics from the Italian population, the KAB framework works. Based on these results, it is necessary to think about and develop an error prevention strategy that overcomes cultural limitations. This concept must be extended to the greatest number of countries worldwide and it is necessary that in every country a validated tool is available to perform an evaluation of ME risks linked to KAB.

Based on this consideration, scores for the behavior prevention scale, which were consistently higher, indicated good knowledge of the behavior that could lead to a medication safety process. In this study, the sample had a lower score related to the behavior scale compared to those previous studied [17,19], and this was due to handwashing before drug preparation and administration, but its value was greater than 4.

Basically, according to Thoma et al. [22], there is a great difference between developing countries and others in terms of knowledge, attitude and behaviour in MEs prevention only due to cultural and organizational context. Indeed, a study conducted in 2021 [43] also showed that there is little difference in mean levels of knowledge, positive attitudes and correct behavior MEs among an international sample.

### 4.1. Study Limitations

This study has some limitations. As a pilot study aimed overall at the intercultural validation of the questionnaire, the sample enrolled is small; therefore, conclusions are not generalizable to the Iranian nurse population. In addition, due to how the authors administered the instrument (web-survey), an accurate response rate was not evaluable. Data are based only on self-reports of KAB. Furthermore, MEs are undoubtedly important and dangerous for ICU, so nurses working in this environment would have a more remarkable aptitude and awareness than those in other settings. Future studies should explore differences and similarities among all countries.

### 4.2. Implication for Clinical Practice and Future Research

The validation and cultural adaptation of the questionnaire represents the first step for realizing an international study. To our knowledge, this is the only study that explores KAB to prevent medication administration errors in ICU among nurses at an international level and that also attempts to delineate cultural aspects of ME prevention. It demonstrates the importance of investigating how cultural background can influence the perception of MEs in order to identify the most appropriate strategy for prevention [22,44,45,46]. A good practice would be to communicate the strengths and weaknesses of similar or different systems to identify the most effective strategies for optimizing the medication management process and patient safety, also in different countries.

## 5. Conclusions

ME prevention is a priority for all countries. This study shows that the KAB medication administration errors in ICU questionnaire is a reliable and valid instrument to be used within Persian healthcare settings, even if a larger sample is desirable to acquire precise information on the level of knowledge, attitude and behaviors regarding MEs in Iranian nurses. Overall, the results regarding configural factorial invariance showed that the nursing population from Italy and Iran used the same cognitive framework to conceptualize MEs prevention. Although having some limits, the present study not only provides a validated instrument to evaluate the KAB theory in Iran, but it is a valid promotion for other researchers in extending this kind of research, supporting those countries where attention to medical error is still increasing. Moreover, although different questionnaires have often been translated into different languages for use in different cultural groups, there is rarely cross-cultural validation to ensure that scores across different cultural clusters could be meaningfully related.

## Figures and Tables

**Figure 1 healthcare-11-00436-f001:**
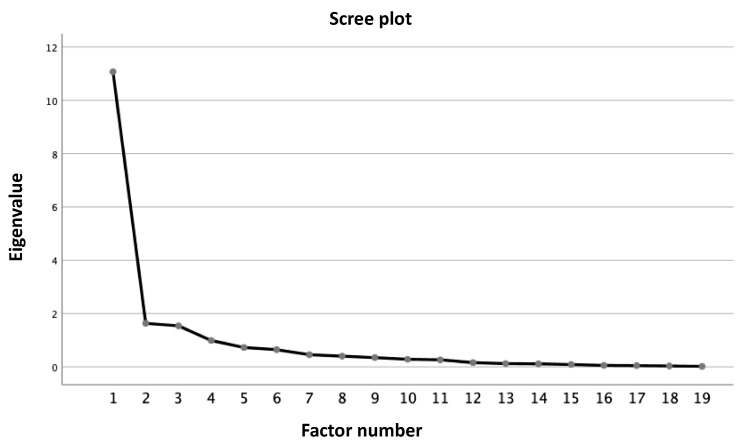
Eigenvalues—Kab theory in medication administration errors in the ICU component.

**Table 1 healthcare-11-00436-t001:** Demographic and personal characteristics of the responding nurses.

	Iranian Sample	Italian Sample Di Muzio et al. [17]
Variables	N (%)	
Age, years		
Mean (range)	36.52	39.9
Gender		
Male	9 (20.5)	169 (31.9)
Female	35 (79.5)	360 (68.1)
Educational qualification		
University degree in nursing	44 (100)	301 (56.9)
Non-university qualification	0	228 (43.1)
Postgraduate training courses		
Master courses	3 (6.8)	144 (88.3)
Other	41 (93.2)	19 (11.7)
Years of work		
Mean (range)	10.98	13.10
Medication management training in university degree		
No	19 (43.2)	31 (5.9)
Yes	25 (56.8)	498 (94.1)
Medication management training in post-degree university courses		
No	19 (43.2)	126 (37.6)
Yes	25 (56.8)	209 (62.4)
English language		
Very low	7 (15.9)	100 (18.9)
Low	13 (29.5)	164 (31.0)
Good	22 (50.0)	90 (17.0)
Very good	2 (4.5)	23 (4.3)
Having the Internet at Work		
No	5 (11.4)	146 (27.7)
Yes	39 (88.6)	382 (72.3)
Having the library access at work		
No	23 (52.3)	146 (27.7)
Yes	21 (47.7)	382 (72.3)
The number of hours per week dedicated to the learning		
<1	38 (86.5)	332 (62.8)
2–5	6 (13.6)	164 (31.0)
6–10	-	24 (4.5)
>10	-	9 (1.7)

**Table 2 healthcare-11-00436-t002:** Exploratory Factor Analysis Results (*n* = 44) and Internal consistency results for the Iranian context.

Item	M	SD	CITC	Factors and Factor Loadings		
				1	2	3
** *Factor 1: knowledge about medication errors prevention* **						
Dosage calculus of intravenous drug reduces preparation errors	4.02	0.849	0.601	0.567	0.223	0.22
Computerized provide order entry system (CPOE) reduce errors during the preparation’s phase	4.05	0.861	0.684	0.627	0.272	0.237
Provision of pre-packaged by the pharmacy reduces medication errors risk	4.09	0.984	0.746	0.778	0.193	0.269
Availability of informative protocols, posters and brochures in the wards, promotes the decrease of the error risk	4.09	0.884	0.841	0.758	0.472	0.2
Assistance of a pharmacist during drug preparation reduces the error risk	4.16	0.914	0.778	0.762	0.315	0.222
Alarm noises and ward emergencies may cause distractions during drugs preparation and administration	4.09	0.960	0.789	0.806	0.307	0.208
Workload (double shifts, extra time) contributes to pharmacological therapy errors	4.34	0.963	0.765	0.666	0.171	0.485
Following the 8 R rule (right patient, right medication, right dose, right route, right time, right documentation, right reason, right response) reduces errors	4.05	1.056	0.675	0.445	0.453	0.292
** *Factor 2: Attitude to medication errors prevention* **						
Ongoing and specific training on safe management of IV drug could reduce the risk of errors	2.80	0.408	0.557	0.23	0.405	0.405
Awareness of the prevention of the errors and management of the clinical risk could reduce the errors during the preparation and administration phases of the drugs	2.77	0.476	0.845	0.56	0.582	0.329
The motivation of the workers can improve their professional performance during the whole medication process	2.89	0.387	0.65	0.282	0.555	0.347
For a secure management of the entire managing process of IV drugs, some authoritative guidelines drawn up taking into account the available scientific evidence are necessary	2.82	0.446	0.669	0.295	0.737	0.201
Protocols/guidelines/procedure can affect professional behaviour, ensuring proper management of therapeutic process	2.77	0.476	0.747	0.333	0.806	0.236
Clinical skills about safe management of drug therapy should be regularly evaluated	2.80	0.462	0.716	0.228	0.899	0.237
Medication errors should be reported in order to become an opportunity to improve the care service	2.80	0.462	0.752	0.314	0.874	0.215
** *Factor 3: Behaviour to medication errors prevention* **						
Hand-washing is necessary before the drug preparation and administration	4.23	0.743	0.803	0.386	0.265	0.85
A check of vital signs before and after the vasoactive drug administration (dopamine, dobutamine, nitroglycerine, etc) reduces complications	4.32	0.674	0.673	0.235	0.18	0.892
Respecting the speed of infusion of the IV administrated solutions (such as chemotherapy, antibiotics, amines, heparin, etc) reduces errors	4.36	0.780	0.736	0.296	0.292	0.807
Before administration, it is necessary to perform a double check to verify the right correspondence among prescription, preparation and administration of the IV drug	4.45	0.627	0.749	0.279	0.443	0.677
Eingevalue				11.070	1.634	1.540
Percentage of variance explained				25.950	25.121	20.250

**Table 3 healthcare-11-00436-t003:** Confirmatory factor analysis and measurement equivalence of KAB theory in medication administration errors in ICU questionnaire across Italian nurses (1) and Iranian nurses (2): 1 vs. 2.

Scale	χ^2^	CFI	TLI	RMSEA	SRMR	Wald Test	Δχ2	Δgl	*p*
Knowledge									
Configural invariance	39.552 (*p* = 0.07) 14.285 vs. 25.464	0.939	0.909	0.094 (0.00–0.158)	0.072	14.285 df: 7—*p* < 0.05	-	-	-
Metric invariance	52.999 (*p* = 0.02) 25.324 vs. 27.675	0.906	0.887	0.105 (0.037–0.160)	0.177	52.514 df: 7—*p* < 0.001	13.447	7	0.06
Scalar invariance	96.745 (*p* < 0.001) 57.252 vs. 39.493	0.713	0.713	0.167 (0.124–0.211)	0.235	64.977 df: 7—*p* < 0.001		43.746	*p* < 0.001
Strict invariance	192.964 (*p* < 0.001) 56.377 vs. 136.588	0.244	0.352	0.251 (0.251–2.89)	0.546	-	96.219	7	*p* < 0.001
Attitude									
Configural invariance	79.116 (*p* < 0.001) 40.594 vs. 38.522	0.904	0.856	0.079 (0.059–0.100)	0.044	59.294 df: 7—*p* < 0.001	-	-	-
Metric invariance	154.199 (*p* < 0.001) 48.802 vs. 105–397	0.776	0.731	0.108 (0.91–0.126)	0.235	10.674 df: 7—*p* = 0.15	75.083	7	*p* < 0.001
Scalar invariance	96.745 (*p* < 0.001) 57.252 vs. 39.439	0.713	0.713	0.167 (0.124–0.211)	0.235	64.977 df: 7—*p* = 0.15	57.454	7	*p* < 0.001
Strict invariance	164.546 (*p* < 0.001) 49.798 vs. 114.748	0.769	0.769	0.100 (−084–0.116)	0.245	-	67.802	7	*p* < 0.001
Behaviour									
Configural invariance	17.519 (*p* = 0.06) 9.409 vs. 8.110	0.979	0.958	0.127 (0.00–0.224)	0.033	0.727 df: 5—*p* = 0.98	-	-	-
Metric invariance	18.251 (*p* < 0.05) 9.811 vs. 8.439	0.991	0.988	0.068 (0.00–0.162)	0.064	19.926 df: 5—*p* = 0.0013	0.732	5	0.98
Scalar invariance	36.726 (*p* = 0.0126) 22.353 vs. 14.373	0.954	0.954	0.134 (0.061–0.202)	0.126	26.809 df: 5—*p* < 0.001	18.475	5	*p* < 0.001
Strict invariance	76.680 (*p* < 0.001) 27.637 vs. 49.043	0.853	0.886	0.211 (0.158–0.265)	0.453	-	39.954	5	*p* < 0.001

## Data Availability

Data available on request due to restrictions privacy and ethical issues.

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
