# Peer review of "Knowledge, Attitude, and Behaviour with Regard to Medication Errors in Intravenous Therapy: A Cross-Cultural Pilot Study"

_healthcare, 2023, doi:10.3390/healthcare11030436_

Round 1

Reviewer 1 Report

Dear Editor,

General Comments

Many thanks for inviting me to review this paper. This study investigated the validation of a questionnaire investigating medication errors. I write my suggestions below.

I believe this study somehow does not align with the scope of the journal. Healthcare is a highly reputable academic journal and has a distinguished audience. And its’ audience deserve high-quality and exquisite publications.

I believe the novelty of the study is questionable. Why a cross-cultural adaptation should be important for international audience? The authors should explain and convince the readers and the reviewers for the importance of their work.

On the other hand, the sampling of the study is rather limited. The ME are important and dangerous for ICU hence the nurses working in this environment would have increase attitude and awareness. Maybe this situation could be added as limitation.

The ethical integrity of this article is questionable.

There are some major limitations of the study. I believe this study is not significant enough to publish Healthcare.

Title

I believe the title is suitable and adheres with the content of the study. However, it is rather long a shorter title would increase readability. Since the questionnaire mainly focused on IV drugs a little heads up within the title might be nice.

Abstract

·         The information about methods section is rather short.

·         I think more information should be given in abstract about methods and conclusion.

·         The take home message should be clarified. I believe detailed outcomes should be underlined. I would like to suggest authors to give some details about the take home message.

·         Keywords: I would like to recommend the adhere MeSH headings.

Introduction

·         I believe the introduction section should be improved. Some sections of the introduction are explanation of questionnaire.

·         Since there are lots of study investigating knowledge, attitude, and behaviour about ME, what is the differences of this study. Why would anyone read this study?

·         The aims are too much in number and complicated.

Methods

·         Study design and sampling method should be elaborated in detail.

·         Did participant sign an informed consent? According to the ethical  authorities the information leaflet should be kept by investigator and participant both. “Consent was implied by the return of a completed questionnaire or by the online completion of the web survey.” This sentence simply is not enough. This situation would lead the articles’ ethical integrity questioned.

·         The authors preferred to use STROBE checklist however CROSS or COREQ checklist would be more useful in terms of their study design.

·         Since a cross-cultural adaptation is made how did the translational sectioned handled?

·         Was there any expert panel to evaluate the translations.

·         How did the translation and cross-cultural adaptation handled. Did the authors make any retrospective cognitive interviews for the suitability of the questionnaire? Or any other method to answer the question above.

·         How did sample size calculated? Since only 44 nurses participated a mathematical calculation is required even for a pilot study.

·         Did test-retest applied for the translated questionnaire for to evaluate the reproducibility?

·         Did the normality checked? And which test were used?

·         Data analysis section is well described.

Results

·         I believe the results section is well organized and beneficial.

·         In table 2 I believe Cronbach's Alpha if Item Deleted column is not necessary.

o   Is CITC-corrected item-total correlation real necessary to show in the table?

o   Also, factor loading either should be presented in another table or within the text.

Discussion

·         The discussion should be elaborated in detail. In present manuscript the discussion is not discussed the hypothesis. Instead, it is simply comparing the results with literature. What are the important factors related with the authors results?

·         There are different models and frameworks to investigate behaviour change what are the superiorities of KAB model you used?

·         What is the take home message? The connection of the aims and the results should be elaborated in detail.

·         What are the future directions for further studies will be held in Iran?

References

·         Need to adhere to the guidance for authors. Please adhere the journal guideline especially for reference sections. Some references have DOI number some don’t, there are few references without containing any page number etcetera.

·         It would be better to give up to date references.

Author Response

Dear Reviewer 1,

We highly appreciate the detailed valuable comments of the referees on our manuscript.

The suggestions were really helpful for us, followed all the comments that were addressed point by point in the revised paper. We hope these efforts will be worked. As below, we clarify the points raised; overall in our opinion it is very important underline that the paper not only provides, for the first time, a validated instrument to evaluate the KAB theory in Iran, but it should promote other researchers in extending this kind of research, supporting those countries where the attention to medical error is still increasing. Thank you again for your support.

General Comments

Many thanks for inviting me to review this paper. This study investigated the validation of a questionnaire investigating medication errors. I write my suggestions below. I believe this study somehow does not align with the scope of the journal. Healthcare is a highly reputable academic journal and has a distinguished audience. And its’ audience deserve high-quality and exquisite publications. I believe the novelty of the study is questionable. Why a cross-cultural adaptation should be important for international audience? The authors should explain and convince the readers and the reviewers for the importance of their work.

Thanks for your suggestion. Medication errors are a prevalent phenomenon that may occur at any stage of the medication process in different healthcare settings. The authors aim to provide preliminary cross-cultural validation for studies with larger samples and different populations to improve professional practice and patient safety in every country. Moreover, according to Thoma et al. [22], the construct of medication errors, their incidence and etiological factors, need to be analyzed also with regard to the personnel cultural background. For instance, some studies report common features in the distribution and type of medication errors between the Italian and Iranian contexts [12-17, 24, 25]. However, the analysis of the studies' methods, settings and cultural backgrounds reveals a great difference that may compromise the comparison. Finally, the paper not only provides, for the first time, a validated instrument to evaluate the KAB theory in Iran, but it should promote other researchers in extending this kind of research, supporting those countries where the attention to medical error is still increasing. This concept was highlighted in both the abstract and conclusion sections.

On the other hand, the sampling of the study is rather limited.

Thank you for your precious comment. This paper report the first step of a larger project: only the translation and intercultural validation of the tool. We have better clarified this issue in the paper. According to Lowe et al (2019) “the pilot studies was used to evaluate the adequacy of planned methods and procedures”. In other words, we known that the instrument was valid among Italian nurses, but we did not know if the instrument could be valid also for Persian nurses. Based on that, the tool was administered to a small sample of nurses to assess the preliminary results, of validation, in a pilot study. Subsequently, it will be extended to a larger and more representative sample to provide consistent data.

The ME are important and dangerous for ICU hence the nurses working in this environment would have increase attitude and awareness. Maybe this situation could be added as limitation.

Thanks for your valuable suggestion. In agreement with the reviewer, the text has been improved in the limits section.

The ethical integrity of this article is questionable. There are some major limitations of the study. I believe this study is not significant enough to publish Healthcare.

As mentioned by Thoma et al. [22], the construct of medication errors, their incidence and etiological factors, need to be analyzed also with regard to the personnel cultural background and there is a great difference between different countries. This is the reason why we conceptualized this study, because there is no Persian tool aiming to assess knowledge, attitude and behavior of nurses about medication errors. Moreover, this study may serve as a valid promotion for other researchers in extending this kind of research, supporting those countries where the attention to medical error is still increasing

Title

I believe the title is suitable and adheres with the content of the study. However, it is rather long a shorter title would increase readability. Since the questionnaire mainly focused on IV drugs a little heads up within the title might be nice.

Thank you for your precious suggestion. The title was changed accordingly: Knowledge, attitude, and behaviour about medication errors in intravenous therapy in Iran: a cross-cultural pilot study.

Abstract

The information about methods section is rather short.

Thanks! The text was changed as follows: A cross-cultural adaptation of an instrument according to the Checklist for reporting of survey studies (CROSS) was performed. The convenience sample was made up of 529 Iranian and Italian registered nurses working in ICU. An exploratory factor analysis was performed, and reliability was assessed. A multi-group confirmatory factor analysis was conducted to test the measurement invariance. Ethical approval was obtained.

I think more information should be given in abstract about methods and conclusion. The take home message should be clarified. I believe detailed outcomes should be underlined. I would like to suggest authors to give some details about the take home message.

            Thanks for the valuable suggestion. We modified the conclusion of the abstract reporting the take home final message: Findings from this preliminary translation and cross-cultural validation confirm that the ques-tionnaire is a reliable and valid instrument within Persian healthcare settings. Moreover, these findings suggest that Italian and Persian nurses used an identical cognitive framework or mental model when thinking about medication errors prevention. The paper not only provides, for the first time, a validated instrument to evaluate the KAB theory in Iran, but it should promote other researchers in extending this kind of research, supporting those countries where the attention to medical error is still increasing.

Keywords: I would like to recommend the adhere MeSH headings.

Thank for your suggestion. According to the MeSH headings, the keyword “cross cultural comparison” has been corrected.

Introduction

I believe the introduction section should be improved. Some sections of the introduction are explanation of questionnaire.

Thanks for the valuable suggestion. We described the questionnaire that was previously used, anyway the description was reduced and text improved.

Since there are lots of study investigating knowledge, attitude, and behaviour about ME, what is the differences of this study. Why would anyone read this study?

Thank you for your comment. The present study aims to translate and validate the KAB theory medication errors questionnaire from English to Persian. To our knowledge, this is the only study that explores KAB to prevent MEs among nurses at an international level and that also attempts to delineate cultural aspects of MEs prevention. It is demonstrated the need of investigating how cultural background can influence the perception of MEs in order to identify the most appropriate strategy of prevention. As previously reported, the paper not only provides, for the first time, a validated instrument to evaluate the KAB theory in Iran, but it should promote other researchers in extending this kind of research, supporting those countries where the attention to medical error is still increasing. In fact, although often different questionnaires have been translated into different languages for use in different cultural groups, rarely there is a cross-culturally validation to ensure that scores across different cultural clusters could be meaningfully related.

The aims are too much in number and complicated.

Thank you for your precious review. The text has been change also according to the suggestion of referee n.1: one aim and two object were identified as follow.

…the present study aims to translate and validate the KAB theory in medication administration errors in ICU questionnaire from English to Persian. Furthermore, two main objectives of the project were: performing a pilot study among Iranian nurses using the translated questionnaire and carrying out a cultural measurement of the KAB theory in medication administration errors in ICU questionnaire across two groups of Italian and Iranian populations...

Methods

Study design and sampling method should be elaborated in detail. Did participant sign an informed consent? According to the ethical  authorities the information leaflet should be kept by investigator and participant both. “Consent was implied by the return of a completed questionnaire or by the online completion of the web survey.” This sentence simply is not enough. This situation would lead the articles’ ethical integrity questioned.

Thank you for your precious review. We modified that in the latest version of our manuscript.

The authors preferred to use STROBE checklist however CROSS or COREQ checklist would be more useful in terms of their study design.

Thank you for your precious review. We used CROSS checklist in the latest version of our manuscript.

Since a cross-cultural adaptation is made how did the translational sectioned handled? Was there any expert panel to evaluate the translations.

Thank you for your precious review. The English version of the KAB theory in medication administration errors in ICU questionnaire was used as a starting point in the translation process. In the translation process, the authors used the recommendations written by the World Health Organization [26] and Beaton et al. [27]. First, two Iranian investigators translated the English versions of the instrument to Persian. To confirm if the Persian translations were reliable, an expert translated them back to English. Afterward, a panel of two expert (fluent in English and Persian) evaluated and compared the retroversion with the original English versions and confirmed their accuracy.

How did the translation and cross-cultural adaptation handled? Did the authors make any retrospective cognitive interviews for the suitability of the questionnaire? Or any other method to answer the question above.

Thank you for your precious review. No retrospective cognitive interviews for the suitability of the questionnaire were conduct. However, at the end of the this pilot study each participant could send us feedback related to understanding or suggestion about the survey. No feedbacks were reported.

How did sample size calculated? Since only 44 nurses participated a mathematical calculation is required even for a pilot study.

Thank you for your precious review. 44 nurses were the total number of nurses working in this setting. For factorial analysis, the Kaiser-Meyer-Olkin (KMO) Measure of Sampling Adequacy was used.

Did test-retest applied for the translated questionnaire for to evaluate the reproducibility?

Again we apologize for the lack of clarity. We have foreseen a paragraph 2.3 Persian validation study: The English version of the KAB theory in the medication errors questionnaire was used as a starting point in the translation process. In the translation process, the authors used the recommendations written by the World Health Organization [26] and Beaton et al. [27]. First, two Iranian investigators translated the English versions of the instrument to Persian. To confirm if the Persian translations were reliable, an expert translated them back to English. Afterward, a panel of two expert (fluent in English and Persian) evaluated and compared the retroversion with the original English versions and confirmed their accuracy.

Did the normality checked? And which test were used? Data analysis section is well described.

Thanks for your comment! The normality of data wasn’t checked because it is not a step for exploratory factor analysis.

Results

I believe the results section is well organized and beneficial. In table 2 I believe Cronbach's Alpha if Item Deleted column is not necessary. Is CITC-corrected item-total correlation real necessary to show in the table? Also, factor loading either should be presented in another table or within the text.

Thank you for your precious review. We modified that in the latest version of our manuscript.

Discussion

The discussion should be elaborated in detail. In present manuscript the discussion is not discussed the hypothesis. Instead, it is simply comparing the results with literature. What are the important factors related with the authors results?

Thank you for your precious review. The discussion was improved accordingly.

There are different models and frameworks to investigate behaviour change what are the superiorities of KAB model you used?

Thank you for your precious review. We have discussed briefly this issue in the text, by the comparison with other scientific experiences.

What is the take home message? The connection of the aims and the results should be elaborated in detail. What are the future directions for further studies will be held in Iran?

Thank you for your precious review. Overall, the paper not only provides, for the first time, a validated instrument to evaluate the KAB theory in Iran, but it should promote other researchers in extending this kind of research, supporting those countries where the attention to medical error is still increasing. Moreover, although often different questionnaires have been translated into different languages for use in different cultural groups, rarely there is a cross-culturally validation to ensure that scores across different cultural clusters could be meaningfully related. This concept was highlighted in both the abstract and conclusion sections.

References

Need to adhere to the guidance for authors. Please adhere the journal guideline especially for reference sections. Some references have DOI number some don’t, there are few references without containing any page number etcetera.

Thanks for your kind review. Reference section has been changed accordingly with reviewer.

THANK YOU FOR YOUR TIME

Reviewer 2 Report

Here they are comments:

The first thing that attaract the attention is the long list of authors for this pilot study. There should be a clear reason for how all these authors have been involved in this research conducted in another country. 

Line 31, the research aim seems to be incomplete. What this it mean to say ''to conduct a linguistic translation and a validation ..'' Also, it does not match the article title as a cross-sectional study.

The research must have only one aim, but it can have several objectives.

The introduction should contain details and statistics of errors in the setting in which this study has been conducted.

Line 65, your claim on the lack of studies on nurses' knowldge and attitude etc is wrong. Here you can find many studies:

https://scholar.google.com/scholar?hl=en&as_sdt=0%2C5&q=nursing+professionals%27+knowledge%2C+attitude%2C+and+behaviour+in+the+preparation+and+administration+of+medication&btnG=

The citation [17] to which you make for introducing the data collection tool in this study is about Knowledge, behaviours, training and attitudes of nurses during preparation and administration of intravenous medications in intensive care units (ICU). Why do you call this questionnaire ''KAB theory in medication errors questionnaire''? This tool as you cited is just about medication administartion in the ICU and can not be generalised to medication errors overall.

Why do you call this study as a pilot? Based on what reason?

Line 97, why only 44 nurses in Iran compared to 529 nurses in Italy? How did you reach the sample size?

Line 98, the correct name in English is 'Tehran'.

Line 99, the process of sampling and recruitment of the Iranian nurses should be described with all details.

Line 100, why the inclusion criteria for the study is so wide? Why did not you concentrate on ICU nurses as the focus of the tool?

bring one overal method to your study and then divide it to two objectives.

You must present the methods section based on your research aim and objectives.

All processes with the psychometric properties of the instrument have been stated under the data analysis, but they should be provided separately.

The description of results are very statistical. You should also present and describe the results for readers who may have not enough knowledge of statistics.

The applicability of the instrument for nursing education and future research is needed. 

Author Response

Dear Reviewer 2,

We highly appreciate the detailed valuable comments of the referees on our manuscript.

The suggestions were really helpful for us, followed all the comments that were addressed point by point in the revised paper. We hope these efforts will be worked. As below, we clarify the points raised; overall in our opinion it is very important underline that the paper not only provides, for the first time, a validated instrument to evaluate the KAB theory in Iran, but it should promote other researchers in extending this kind of research, supporting those countries where the attention to medical error is still increasing. Thank you again for your support

The first thing that attaract the attention is the long list of authors for this pilot study. There should be a clear reason for how all these authors have been involved in this research conducted in another country. 

Thank you for your comment; the authors are aware of its meaning. The following authors were involved in obtaining a reliable and sensitive tool: authors of the original Italian instrument; Italian and Persian language experts for back-translation; experts in quantitative research and statistics; supervisors to resolve disagreements. In particular, the tool translation and back translation process alone, involved five people as described in paragraph 2.3. Persian validation study.

Line 31, the research aim seems to be incomplete. What this it mean to say ''to conduct a linguistic translation and a validation ..'' Also, it does not match the article title as a cross-sectional study.

Thanks for your kind comment. The text has been changed accordingly: A pilot study was conducted for the Persian cross-cultural validation of the Italian questionnaire on knowledge, attitude, and behavior theory regarding medication errors.

The research must have only one aim, but it can have several objectives.

Thanks for your kind comment, we have better identified the aim and the objectives. The text has been changed accordingly: …….the present study aims to translate and validate the KAB theory medication errors questionnaire from English to Persian. Furthermore, two main objectives of the project were: performing a pilot study among Iranian nurses using the translated question-naire and carrying out a cultural measurement of the KAB theory in the medication errors questionnaire across two groups of Italian and Iranian populations…….

The introduction should contain details and statistics of errors in the setting in which this study has been conducted.

Thanks for your kind comment. We improved the introduction section by detailing the study context.

Line 65, your claim on the lack of studies on nurses' knowldge and attitude etc is wrong. Here you can find many studies:

https://scholar.google.com/scholar?hl=en&as_sdt=0%2C5&q=nursing+professionals%27+knowledge%2C+attitude%2C+and+behaviour+in+the+preparation+and+administration+of+medication&btnG=

Thanks! According to your suggestion the text has been changed in: Although increasing, the literature focused on nursing professionals' knowledge, attitude, and behaviour (KAB theory) in the preparation and administration of medication do not cover the majority of the countries [17].

The citation [17] to which you make for introducing the data collection tool in this study is about Knowledge, behaviours, training and attitudes of nurses during preparation and administration of intravenous medications in intensive care units (ICU). Why do you call this questionnaire ''KAB theory in medication errors questionnaire''? This tool as you cited is just about medication administartion in the ICU and can not be generalised to medication errors overall.

Thank, we agree. The name was changed in “KAB theory in medication administration errors in ICU questionnaire” along the paper

Why do you call this study as a pilot? Based on what reason?

Thank you for your precious comment. This paper report the first step of a larger project: only the translation and intercultural validation of the tool. Thanks to your previous comments, we have better clarified this issue in the paper. According to Lowe et al (2019) “the pilot studies was used to evaluate the adequacy of planned methods and procedures”. In other words, we known that the instrument was valid among Italian nurses, but we did not know if the instrument could be valid also for Persian nurses. Based on that, the tool was administered to a small sample of nurses to assess the preliminary results, of validation, in a pilot study. Subsequently, it will be extended to a larger and more representative sample to provide consistent data.

Line 97, why only 44 nurses in Iran compared to 529 nurses in Italy? How did you reach the sample size?

Thanks for the valuable comment. As described in paragraph "2.2. Iranian sample", convenience sampling was used. Only the nurses in the hospital hosting the research who could speak and read Persian fluently were involved in the study. Overall, 44 nurses were the total number of nurses working in this setting.

Line 98, the correct name in English is 'Tehran'.

            Done, thanks! The text has been changed accordingly.

Line 99, the process of sampling and recruitment of the Iranian nurses should be described with all details.

            Thank you for your kind revision. We agree with the Reviewer and the text has been changed in: The sample was selected through a convenience sample. Overall, all the Persian registered nurses who work in ICUs in hospital of Tehran, Iran (n.44) completed a self-administered questionnaire. Researchers sent an email to all nurses working in ICU. The consent was obtained before starting the data collection. Based on a web survey, data was collected in February 2019. The inclusion criteria were nurses who worked in the ICU at the hospital hosting the research and could speak and read Persian fluently.

Line 100, why the inclusion criteria for the study is so wide? Why did not you concentrate on ICU nurses as the focus of the tool?

Thanks for your valuable comment. There was a mistake. We modified the text. Only nurses working in ICUs was included in this study.

bring one overal method to your study and then divide it to two objectives.

Thank you for your kind review. We agree with the Reviewer and the text has been changed.

You must present the methods section based on your research aim and objectives.

Thanks! The text has been changed accordingly.

All processes with the psychometric properties of the instrument have been stated under the data analysis, but they should be provided separately.

            Done, thanks! The text has been changed accordingly.

The description of results are very statistical. You should also present and describe the results for readers who may have not enough knowledge of statistics.

Done, thanks! The text has been changed accordingly. We present and describe the results for readers who may have not enough knowledge of statistics into the discuss section.

The applicability of the instrument for nursing education and future research is needed. 

            Thank you for your precious suggestion. The text has been changed with the addiction of a new section:

            4.2. Implication for clinical practice and future research.

THANK YOU FOR YOUR TIME

Round 2

Reviewer 1 Report

Dear Editor,

I would like to thank author for accepting suggesting revision. However due the lack of novelty, etcihal issues, focusing only one small sample size I believe this study is not significant enough to be published in the journal of healthcare.

Regards. 

Author Response

Dear Reviewer 1,

We highly appreciate the detailed valuable comments provided after the evaluation of the manuscript.

The suggestions were really helpful for us, we followed all the comments that were addressed point by point in the revised paper. We hope these efforts will be worked. As below, we clarify the points raised. Thank you again for your support:

Dear Editor, I would like to thank author for accepting suggesting revision. However due the lack of novelty, etcihal issues, focusing only one small sample size I believe this study is not significant enough to be published in the journal of healthcare. Regards. 

Thank you for your valuable comment. The authors are aware of the limitation provided by the small sample that were discussed in the limits section. However, the present pilot study aims to provide a preliminary evaluation of the translation and cross-cultural validation of the tool to be tested in more extensive future studies. The use of not pre-evaluated and unreliable instrument is a serious and irreversible limitation of many studies. The fact that the tools refer to previously used surveys is not sufficient. A validation process must be always reported. Moreover, as mentioned by Thoma et al. [2019], the construct of medication errors, their incidence and etiological factors, need to be analyzed also with regard to the personnel cultural background and there is a great difference between different countries (e.g. Italy vs Iran). This is the reason why we conceptualized this study, because there is no intercultural validated tool aiming to assess knowledge, attitude and behavior of nurses about medication errors.

Thank you very much for your time, the authors

Reviewer 2 Report

The article has been improved, but more amendments are needed. 

As you have divided the methods section using subheadings for ''Exploratory Factor Analysis'' and ''Confirmatory Factor Analysis'', therefore, present the research results under the same subheadings and bring all statistical details and tables for each of them in the results section. It helps readers to follow up the research process and results. Also, add a copy of the instrument as the supplmentary file to this article, and cite it in the text.

Author Response

Dear Reviewer 2,

We highly appreciate the detailed valuable comments provided after the evaluation of the manuscript.

The suggestions were really helpful for us, we followed all the comments that were addressed point by point in the revised paper. We hope these efforts will be worked. As below, we clarify the points raised. Thank you again for your support:

The article has been improved, but more amendments are needed. As you have divided the methods section using subheadings for ''Exploratory Factor Analysis'' and ''Confirmatory Factor Analysis'', therefore, present the research results under the same subheadings and bring all statistical details and tables for each of them in the results section. It helps readers to follow up the research process and results. Also, add a copy of the instrument as the supplementary file to this article, and cite it in the text.

Thanks for your suggestion. In the latest version of our manuscript, result section was divided into: characteristic of the sample; Exploratory Factor Analysis; Confirmatory Factor Analysis and measurement equivalence. Moreover, Table 2 is referred to exploratory factor analysis, while Table 3 was referred to Confirmatory Factor Analysis. A copy of instrument was provided as supplementary file and cited in the text.

Thank you very much for your time, the authors
